# Effects of Excitatory Repetitive Transcranial Magnetic Stimulation of the P3 Point in Chronic Stroke Patients—Case Reports

**DOI:** 10.3390/brainsci8050078

**Published:** 2018-04-28

**Authors:** Ronaldo Luis da Silva, Angela Maria Costa de Souza, Francielly Ferreira Santos, Sueli Toshie Inoue, Johanne Higgins, Victor Frak

**Affiliations:** 1Faculté des Sciences, Université du Québec à Montréal, Montreal, QC H2X 3Y7, Canada; frak.victor@uqam.ca; 2Centro de Reabilitação e de Readaptação Dr Henrique Santillo—CRER, Goiânia, Goiás 74653-230, Brazil; angelamcsouza52@gmail.com (A.M.C.d.S.); francielly.fisio2010@gmail.com (F.F.S.); inoue.sueli@gmail.com (S.T.I.); 3École de Réadaptation, Faculté de Médecine, Université de Montréal, Montreal, QC H3N 1X7, Canada; johanne.higgins@umontreal.ca

**Keywords:** intraparietal sulcus, stroke, rTMS, Fugl-Meyer Assessment, fast frequency TMS, motricity, sensibility, chronic patients

## Abstract

Objective: To evaluate the effects of excitatory repetitive transcranial magnetic stimulation (rTMS) of the international 10–20 system P3 point (intraparietal sulcus region) in chronic patients with a frontal lesion and parietal sparing due to stroke on the impaired upper (UL) and lower limb (LL) as measured by the Fugl-Meyer Assessment (FMA). Methods: Three patients (C1: 49.83/2.75, C2: 53.17/3.83, C3: 63.33/3.08-years-old at stroke/years post-stroke, respectively) received two weeks (five days/week) of rTMS at 10 Hz of P3. A patient was treated in similar conditions with a sham coil (S1: 56.58/4.33). Patients were evaluated before, after, and two months post-treatment (A1, A2, and A3, respectively). Results: For LL, the scores of the motor function subsection of C1 and C3 as well as the sensory function of C2 increased by A2 and remained by A3. For UL, the score of the motor function of C2 and C3 also increased, but the score of C3 decreased by A3. The score of the range of motion subsection of C3 increased by the two follow-up evaluations. Conclusion: This study suggests excitatory rTMS over P3 may be of use for some chronic stroke patients, but these findings need to be verified in a future clinical trial.

## 1. Introduction

Transcranial magnetic stimulation (TMS) is a widely studied tool for the treatment of post-stroke patients [1]. Several studies have obtained promising results in the treatment of depression [2,3], aphasia [4,5,6,7], and pain [8,9,10,11], as well as the improvement of motor function [3,12,13,14,15]. However, the variety of results of TMS with the post-stroke population requires further study. Such studies are predominantly based on the interhemispheric imbalance model [16], stating that the injury of one hemisphere increases the activation of the contralateral hemisphere, which, in turn, exerts a greater inhibition over the injured hemisphere [16,17,18]. The majority of these studies have applied inhibitory repetitive transcranial magnetic stimulation (rTMS) to the intact hemisphere and excitatory rTMS to the injured hemisphere [4,14,16]. However, excitatory stimulation both presents opposite results from inhibitory stimulation and its results tend to be broader and more intense. In contrast, inhibitory stimulation tends to generate changes in a smaller number of cortical centers with a lower intensity [18,19]. Several researchers have applied the excitatory stimulation on the usually inhibited unlesioned hemisphere in patients with aphasia or motor impairments resulting from brain lesions [6,20]; they found similar or more consistent results compared to those obtained by inhibitory stimulation. On the basis of these findings, a positive effect may also be possible with excitatory rTMS in a post-stroke population that is not restricted to the model of inter-hemispheric imbalance.

Studies evaluating the effects of rTMS on motor function have typically used the primary motor cortex as the stimulation site [8,9,12,13,14,15,21]. These studies have obtained good results with acute [14] and chronic patients [8,12,15,21]. However, direct application to the primary motor cortex may restrict the excitatory rTMS effects to the stimulated neurons since the main output of the primary cortex is directed to non-cortical areas, thus reducing the effectiveness of excitatory stimulation [22,23].

The P3 point in the international 10–20 system corresponds to the intraparietal sulcus, a largely gyrified region [24] in the human brain which has been receiving increasing attention from the scientific community as a result of its relevance in sensorimotor integration and in several aspects of motor coordination, such as motor planning [25,26,27], reaching and gripping/grasping [28,29], and online correction [29]. According to Herwig et al. [30], the P3 stimulation may achieve intraparietal sulcus or surrounding regions in the Brodmann areas 7 (BA 7) and 40 (BA 40).

It has been noted that BA 40’s activation intensity is greater in people with long-term motor training [31]. In addition, its earlier activation in the post-stroke acute phase has been correlated with better motor recovery [32]. Both BA 40 and BA 7 have already been related to sequential finger movements [33,34]. Their activation grows from unimanual to bimanual movements as well as from symmetrical to asymmetrical bimanual movements [35]; they also are involved in extracting task-relevant information when different inputs are available [36]. Left BA 7 was also correlated to reaching and grasping in unimanual tasks [37].

As a tertiary cortex, this region is connected to numerous other regions [25,38]; therefore, its stimulation could lead to a broader influence on motor function. However, this region is also usually damaged in more extensive strokes which involve the middle cerebral artery. A stroke that spared the lower trunk or the parietal branch of the middle cerebral artery would preserve the intraparietal sulcus and its surrounding regions [39,40]. Excitatory magnetic stimulation of this spared region could provide information about the effect of positive stimulation of a spared area on originally connected injured areas within the same hemisphere. In particular, the excitatory stimulation of the P3 point could have positive effects on the motor and sensory functions on the basis of findings already described associated with the stimulated region.

## 2. Materials and Methods

### 2.1. Ethics Statement

The project was approved by the Université du Québec à Montréal, Canada. Ethical approval was obtained from the UNICEUB Research Ethics Committee (CEP-UNICEUB), Brasília, Brazil—report No. 2.044.460/17.

### 2.2. Subjects

Participants were selected from a comprehensive analysis of the medical records of patients seen at Dr. Henrique Santillo Rehabilitation and Readaptation Center–CRER’s outpatient clinic, from January to October 2017 in Goiânia, Brazil. To be included in the study, patients had to have a diagnosis of a first-ever left-hemisphere stroke which involved the middle cerebral artery two to five years prior to the study. Additionally, the parietal lobe had to have been spared by the stroke. Analysis of the lesion extension and parietal sparing was based on imaging examinations by the patient’s neurologist and the research team. Patients had to be between 40 and 70 years old and consistently right-handed prior to stroke according to the Edinburgh Inventory [41]. In addition, the following exclusion factors were considered: neurodegenerative diseases, moderate to severe musculoskeletal disorders previous to stroke, psychiatric disorders, uncorrected or stroke-related visual impairments, diabetes mellitus, and any contraindications for TMS procedures. Eligible participants agreed to participate in the study by signing the informed consent form. A personal companion was present at the presentation of the research and the signing of the informed consent form.

### 2.3. Evaluations

Patients were evaluated with the Fugl-Meyer Assessment (FMA) before the treatment (A1). An occupational therapist evaluated the upper limb, and a physical therapist evaluated the lower limb. These assessments were repeated at the end of the treatment (A2) and two months after A2 (A3). Evaluations were administered in the morning, in the same room, by the same professionals.

### 2.4. rTMS

To determine each participant’s resting motor threshold (RMT), the coil was positioned tangentially on the scalp with the handle directed upward and posteriorly at 45° to the frontal plane, nearly parallel to the central sulcus. Single TMS pulses were applied to the participant’s left M1 on the C3 point of the international 10–20 system. RMT was defined as the lowest level of machine output that elicited three twitches in the first dorsal interosseous of six consecutive TMS pulses [42]. Repetitive TMS was performed with a Neurosoft stimulator with a 76 mm figure-of-eight coil on the P3 point of international 10–20 system, which predominantly refers to the intraparietal sulcus in the left hemisphere [30] where the anterior intraparietal area is located. Figure 1 shows the P3 positioning and the stimulated areas according to Herwig et al. [30], illustrating the proportionality for each one. We delivered 40 trains of 50 pulses each at 10 Hz and 90% RMT of each individual patient at 25 s intervals, totaling 2000 pulses in a 20 min session for two weeks (five days/week). The 10 Hz frequency was chosen according to international TMS guidelines, which advise that the 10 Hz frequency must be preferably chosen relative to 20 Hz, 15 Hz, and 5 Hz; the other parameters were in accordance with the safety ranges for high-frequency rTMS [1]. Blood pressure was evaluated prior to, immediately after, and five minutes after each rTMS session. The coil was positioned tangentially on the scalp with the handle pointing posteriorly to the base of the neck at 30° relative to the transverse plane. This position follows the positioning described by Koch et al. [43] to better achieve the anterior intraparietal area. Participants lay down their side on a stretcher during stimulation with their head supported for comfort and better positioning of the coil. The sham patient was equally positioned, but the sham coil was unattached to the stimulator, while the active coil was kept near the sham coil to provide the sham auditive stimulation.

## 3. Results

Medical records of patients resulted in the pre-selection of seven patients, four of whom agreed to participate. One patient was randomly chosen to receive sham treatment. Patient 1 (C1—woman) was 49 years old and 2.75 years post-stroke. Patients 2 and 3 (C2 and C3—men) were 53 and 63 years old, with 3.83 and 3.08 years post-stroke, respectively. The patient who received the sham treatment (S1—man) was 56 years old and 4.33 years post-stroke. Figure 2 shows the spared intraparietal sulcus and the lesioned M1 of each participant. The images were performed as part of the medical monitoring of each patient and outside the institution where this study took place, therefore, without the purpose of serving as the basis for scientific research. The Fugl-Meyer Assessment (FMA) scores are found in Table 1.

Patient C1 increased six points on the FMA lower limb motor function subsection after rTMS treatment, and this increase was still present two months after the end of the treatment when the score reached the maximum value. She gained two points on the pain subsection by A2 and reached the maximum value by A3; she also gained two points on the sensory function subsection by A3. She was the only patient to present some idiopathic chronic pain after stroke. Although the patient reported some difficulty in performing activities of daily living (ADLs) with the right hand, FMA was unable to find any impairment in motor function subsection, since she reached the highest score at baseline. Patient minimally decreased the upper limb pain score by A3, indicating an increase in hand pain level.

Patient C2 increased his score on the FMA lower limb sensory function subsection by six points, reaching the maximum score for this subsection, and this increase remained by A3. Motor function and range of motion subsections minimally fluctuated by A2 and A3. He gained five points by A2 on the upper limb motor function subsection, but this gain was lost by A3. No changes were observed in the other subsections.

Patient C3 presented the lowest scores for lower limb motor function subsection at baseline, and he increased its score by four points by A2. This gain remained by A3. He also gained a single point for the sensory function by A2 that remained by A3. His score on the upper limb motor subsection was also the lowest in the group, indicating severe hemiparesis. By the end of the treatment, he regained the ability to hold an object with the hand and release it when solicited, granting an additional four points by A2. This ability was still present by A3. The range of motion subsection presented a discrete increase by A2 that reached six points compared to A1 by A3 and an increase of two points on the pain subsection by A3. These gains corresponded to the hand and wrist.

Patient S1 only presented a single point fluctuation in lower limb sensory function and pain subsections and no changes in upper limb subsections by A2; therefore, he did not participate in A3.

Score variations by subsection for lower limb and upper limb can be found in Figure 3 and Figure 4, respectively.

## 4. Discussion

This study aimed to investigate the effects of the excitatory magnetic stimulation of the P3 point on the all FMA subsections scores of the impaired lower limb and upper limb in three chronic stroke patients whose intraparietal sulcus region was spared by the middle cerebral artery stroke. We found an increase in motor function, sensory function, and pain level scores (which indicates a reduction in pain level according to FMA) for the affected lower and upper extremities, suggesting that the rTMS of this spared region could yield wide-ranging benefits.

### 4.1. Lower Limb

Both patients C1 and C3 had an improvement of their lower limb motor function score as assessed by the FMA. Patient C1 increased her score six points by A2 and gained one more point by A3, reaching the maximal score on this FMA subsection. Pandian et al. [44] found that a six-point change in the motor function subsection in chronic stroke patients is clinically important; therefore, from a clinical perspective, her score was significantly changed from baseline to post-treatment evaluations. She also presented a progressive increase on the sensory subsection and a reduction of pain. For patient C3, the improvement of his motor function score did not reach the minimal clinically important difference indicated by Pandian et al. [44]; this improvement was accompanied by a slight increase in sensory function. Although these variation values were low, they mirrored the motor and sensory gains observed in patient C1.

Patient C2 also showed important gains in the sensory function of the lower limb, but there are no studies indicating a clinically important minimal difference for sensory function. Although he showed the greatest gains in sensory function among the three treated patients, motor function variation did not mimic these gains. This may be a result of the variability of effects of the stimulation or, more likely, the patient’s specific central compromises. Because the patient started the study with 29 points out of a maximum of 34 points, a gain of five points would raise him to the normal range, without allowing him to reach the six points necessary for clinical significance. Accordingly, the best condition of his right lower limb might explain the difference between him and the other patients.

Several studies have elucidated the relevance of sensory function to motor performance after a stroke [46,47,48,49]. A rehabilitation that aims to improve sensory functions tends to produce better results [47,48] because sensory integration is the base of the elaboration and structure of movement [46]. In this study, the excitatory stimulation of the P3 point increased the sensory function score of the three tread participants, reaching the subsection maximum score for patients C1 and C2. The combined gains in sensory and motor functions made a stimulation model even more beneficial to the patient because these functions are interrelated; an improvement in one area may directly impact the other. Sensory and motor rehabilitation therapies could benefit from these gains obtained from stimulation in chronic stroke patient care. Our findings in these cases suggest that the excitatory rTMS of the P3 point may be beneficial to the lower limbs both for motor function and for sensory function in this stroke population.

### 4.2. Upper Limb

Patient C3 presented the important gain of an active palmar grasp, which he was unable to perform by A1. Hand and wrist gains accounted for the increase in motor function and range of motion subsections of the FMA. These gains were found at the end of the treatment and reached even greater values by the two month evaluation (A3), when a slight increase in pain reduction was also found. Together, these changes reflected both a reduction in muscle tonus and a better voluntary motor control.

Patient C2 had an important gain in motor function subsection at the post-treatment evaluation, but this score reduced by the two month evaluation.

According to Page and Hulk [45], the clinically important difference for grasping ability is 4.25 points; however, for the general function of the upper limb, it is 5.25. Thus, the values achieved both by patient C2 in A2 and patient C3 in A2 and A3 are clinically important.

### 4.3. Sensory Function

Both the inferior parietal lobe and intraparietal sulcus are strongly connected to the frontal cortex [50]. In particular, the anterior intraparietal area, which corresponds to the anterior portion of the intraparietal sulcus, is described as an important node for grasping processing [51,52] due to its connections with parietal and frontal areas; however, to our knowledge, no study has linked the intraparietal sulcus and surrounding regions to the lower limb motor function. Connections between the parietal cortex and the frontal cortex in a parietal-premotor network are key for sensory-motor control [25]. This network is compounded by several pathways related to reaching, grasping, body imaging, spatial processing, and diverse modalities of sensory input which are linked to different portions of the intraparietal sulcus [21,25]. Therefore, the region stimulated in our study may be related to self-image construction by means of sensorimotor input [26,53]. The activation of the parietal cortex was correlated with a better sensory discrimination in chronic stroke patients [54]. Here, we found that the excitatory stimulation of the P3 point area improved the lower limb FMA sensory function score for the treated patients in different values, suggesting that the excitatory stimulation of P3 could improve the sensory processing of the lower limb. However, an analysis of these results needs to consider the intrinsic limitations of case reports, such as the influence of natural variation among subjects as well as the influence of other limitations described below. Two patients obtained the maximum score prior to the treatment and the third patient did not change his score after the treatment. Therefore, we cannot evaluate to what extent excitatory stimulation of P3 improves the sensory function.

### 4.4. Limitations

The major limitation of our study was the small number of patients. We evaluated 540 medical records in this study, which set this condition prevalence at slightly over one percent. Although this index may vary in different centers or countries according to the promptness of stroke care assistance, it should still be small. Although it is a relatively rare condition, it highlights the possibility of studying the influence of ipsilaterally applied transcranial magnetic stimulation on a spared area which is closely related to regions affected by stroke.

Although patient C1 had reported difficulties in performing ADLs with the affected upper limb, she obtained the highest score for the motor function in the pre-evaluation. In this case, FMA was not sensitive enough for this patient. To our knowledge, no studies have provided a minimal clinically important score for the FMA sensory function and pain absence subsections. Therefore, it still lacking a parameter to proceed with an integrated analysis of the different subsections of FMA [55]. Although FMA is recommended as a validated tool for primary outcomes in intervention trials [56,57], the lack of methods for individualized and integrated analysis of its subsections reduces its effectiveness. 

Our study used the international 10–20 system to determine the stimulation site. The use of a neuronavigation system and individual structural magnetic resonance imaging could lend greater uniformity to the results, and the replication of the study with this apparatus might confer greater confidence regarding the effects of the P3 excitatory stimulation. Although the P3 stimulation aims to activate the intraparietal sulcus, three different Brodmann areas could be activated according to Herwig et al. [30]. These areas have distinct connections and are engaged in different circuitries, which reduces the possibility of knowing precisely which brain regions are activated by P3 stimulation and, accordingly, which brain regions may have caused the observed clinical changes. The delimitation of the stimulation area could provide a more consistent basis for understanding and developing new projects. However, the international 10–20 system’s ease of application, quality, and low cost make this method a suitable tool for replication [30]. The increased possibility of variation of the stimulation site associated with the international 10–20 system makes this technique more suitable for large samples. The likely variation of the stimulation site limited our conclusions and the possibility of further generalizations. Our results were not uniform, as expected in such a reduced sample size, and the use of the international 10–20 system may have corroborated with this variety. Among the 24 measures evaluated (eight subsections for each treated patient), seven presented an increase from A2 to A3. These measures indicate that the effects of rTMS may have been partially progressive during this follow-up period. Again, caution is needed when considering case reports studies. Data from the two-month follow-up evaluation with the sham patient could serve as a point of comparison for these findings; however, unfortunately, this evaluation was not performed. Nevertheless, hypotheses that might be drawn from our observations with these three treated chronic stroke patients could positively contribute to overall rehabilitation research with stroke patients.

## 5. Conclusions

In these case reports, our findings suggest that the excitatory stimulation of the P3 might increase lower and upper limb Fugl-Meyer Assessment scores in motor and sensory functions, as well as in pain reduction in chronic stroke patients whose intraparietal sulcus and surround regions were spared in a middle cerebral artery stroke. However, as this is a case series of only three patients and one sham, we caution readers when interpreting these findings. These encouraging findings need to be verified in a future study utilizing an appropriate clinical trial design.

## Figures and Tables

**Figure 1 brainsci-08-00078-f001:**
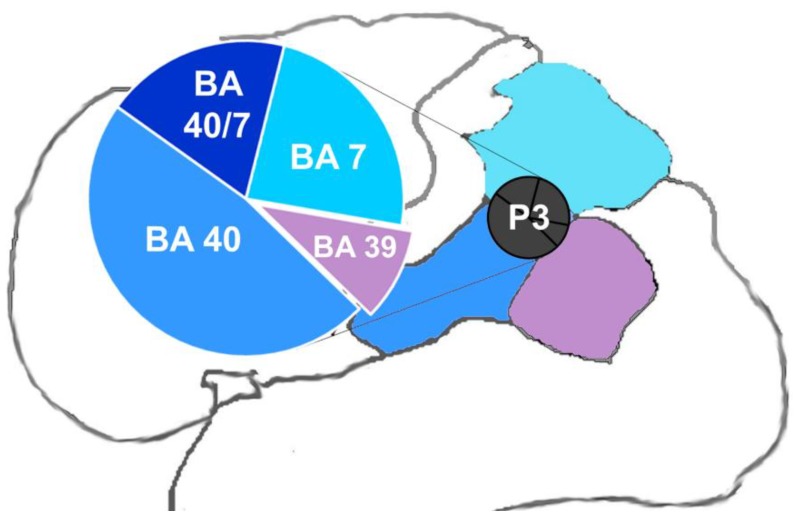
Possible stimulated areas by the international 10–20 system P3 point. The main figure illustrates the P3 positioning in the brain. The highlighted graphic illustrates the probability associated with each Brodmann area according to Herwig et al. [30]. BA 40 = Brodmann area 40 close to the intraparietal sulcus; BA 7 = Brodmann area 7 close to the intraparietal sulcus; BA 40/7 = intraparietal sulcus; BA 39 = Brodmann area 39 close to Brodmann area 40.

**Figure 2 brainsci-08-00078-f002:**
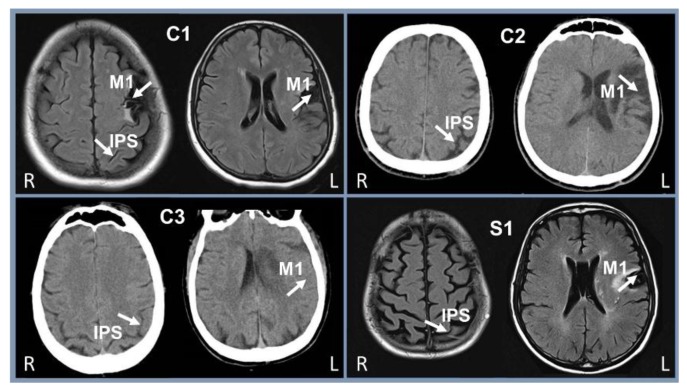
Computed tomography (patients C2 and C3) and magnetic resonance image (patients C1 and S1) showing the spared intraparietal sulcus and the affected primary motor cortex. IPS = intraparietal sulcus; M1 = primary motor cortex; C1-3 = treated patients; S1 = sham patient. L = left side of the brain; R—right side of the brain.

**Figure 3 brainsci-08-00078-f003:**
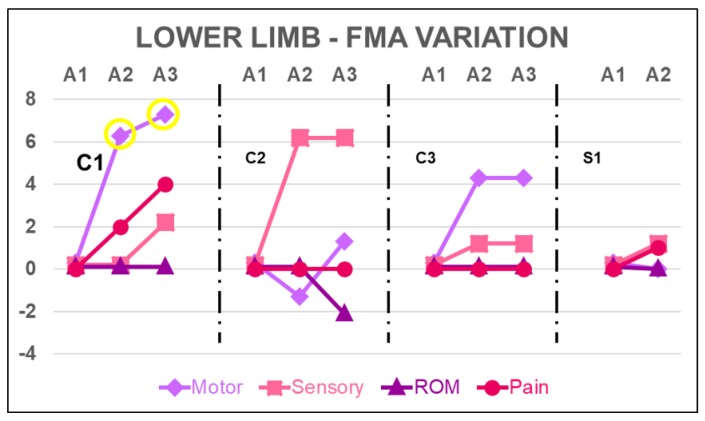
Score variation for lower limb Fugl-Meyer Assessment subsections. Yellow circles indicate minimal clinically important difference for FMA motor subsection according to Pandian et al. [44].

**Figure 4 brainsci-08-00078-f004:**
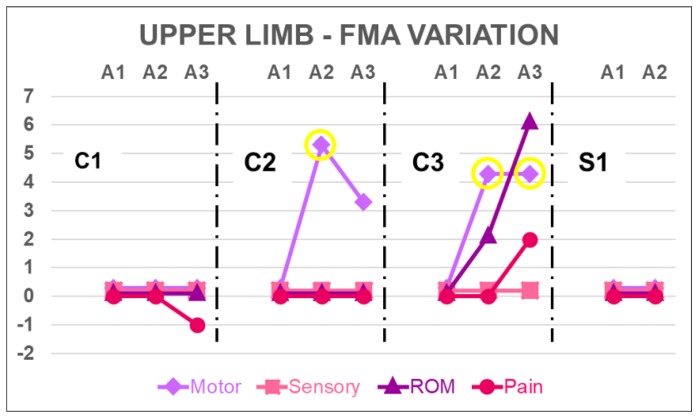
Score variation for upper limb Fugl-Meyer Assessment subsections. Yellow circles indicate clinically important difference for grasping in FMA motor subsection according to Page & Fulk [45].

**Table 1 brainsci-08-00078-t001:** Fugl-Meyer Assessment subsections scores.

			C1			C2			C3			S1	
		Max	A1	A2	A3	A1	A2	A3	A1	A2	A3	A1	A2
**LL-FMA**	motor function	34	27	33	34	29	28	30	17	21	21	18	18
	sensory function	12	10	10	12	6	12	12	9	10	10	10	11
	ROM	20	20	20	20	20	20	18	18	18	18	16	16
	joint pain	20	10	12	14	20	20	20	20	20	20	19	20
**UL-FMA**	motor function	66	66	66	66	13	18	16	4	8	8	2	2
	sensory function	12	12	12	12	12	12	12	6	6	6	6	6
	ROM	24	24	24	24	24	24	24	18	18	24	13	13
	joint pain	24	23	23	22	20	20	20	18	18	20	20	20

C1, C2, C3: treated patients; S1: sham treated patient; max: subsection maximum score; A1: pre-treatment evaluation; A2: post-treatment evaluation; A3: two-month follow-up evaluation; LL-FMA: lower limb Fugl-Meyer Assessment; UL: upper limb Fugl-Meyer Assessment; ROM: range of motion.

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
