# Peer review of "Effects of Excitatory Repetitive Transcranial Magnetic Stimulation of the P3 Point in Chronic Stroke Patients—Case Reports"

_brainsci, 2018, doi:10.3390/brainsci8050078_

Round 1

Reviewer 1 Report

This is interesting work describing a novel stimulation approach for improving limb motor function in chronic stroke patients.

Comments and suggestions:

Title: Need to include the words "case reports" to indicate the small-sample nature of this investigation.

Abstract: (Lines 26-27) Need to tone down the conclusion to reflect the fact that this was an observation from three cases. The language implies too broad of a conclusion.

Introduction: (Lines 41-44) Please check the accuracy of this statement. The provided references [5,19] do not demonstrate the stated results with regards to depression in stroke patients.

(Lines 44-45) Soften your conclusion regarding the implication of these studies. Perhaps use “These studies suggest the possibility that…” in light of the fact that this statement is being made based on preliminary findings with small numbers of patients.

Materials and Methods: (Lines 75-76) Chronic stroke is often defined as being 6 or 12 months post-stroke. Please provide rationale for requiring patients to be specifically 2-5 years post-stroke for this study.

(Lines 99-100) Please provide rationale for stimulating at 90% of resting motor threshold, as this intensity of stimulation is often considered under-powered.

Results: I strongly suggest including a figure showing imaging (MRI/CT) results for the patients that demonstrates lesion location and evidence of spared parietal lobe.

Discussion: (Lines 194-195) Do the authors have any speculations as to why hand and wrist gains were even greater at the two-month evaluation compared to immediately post-treatment for P3?

(Line 218) The statement “Precisely for this reason…” seems illogical. Why does the low prevalence of this condition specifically warrant the use of an excitatory protocol?

(Lines 222-223) Again, the statement “Thus, the model justifies conducting the study even with few patients” does not seem to be validated by the accompanying text.

I suggest re-writing the section from Line 212-223 to improve the logical flow of your arguments. It may well be justified to conduct this study with few patients, but the arguments provided do not seem to support that conclusion as-written.

(Lines 240-243) Concluding that the heterogeneity of the observed results is a strength of the study seems like a stretch. I suggest re-wording. This sentence is rather confusingly worded in general. Perhaps tone down the language regarding the strength of a heterogeneous result and instead point out that these preliminary results suggest that this protocol has the potential to improve a range of impaired functions.

Author Response

Reviewers report

Reviewer 1

This is interesting work describing a novel stimulation approach for improving limb motor function in chronic stroke patients.

     We would like to thank you for you comments. We are sure that they have greatly contributed to the improvement of our article. We hope we have responded appropriately to all of them.

1.   Title: Need to include the words "case reports" to indicate the small-sample nature of this investigation.

     We included this statement (line 4).

2.   Abstract: (Lines 26-27) Need to tone down the conclusion to reflect the fact that this was an observation from three cases. The language implies too broad of a conclusion.

     We reduced the phrase to “In a variable way, AIP excitatory rTMS increased FMA scores in different upper and lower limb subsections of our three treated patients”, eliminating the comparison to M1 stimulation (lines 26-27).

3.   Introduction: (Lines 41-44) Please check the accuracy of this statement. The provided references [5,19] do not demonstrate the stated results with regards to depression in stroke patients.

     We apologize. This statement is correct concerning the aphasia (reference 5 in the text) but not the depression. Since depression courses with inter-hemisphere imbalance without an established lesion as AVC does, it does not fit this statement, and the reference is not correct too. We modified the phrase, changing “depression” for “motricity impairment due to brain lesion”, therefore justifying the 19th reference.

4.    (Lines 44-45) Soften your conclusion regarding the implication of these studies. Perhaps use “These studies suggest the possibility that…” in light of the fact that this statement is being made based on preliminary findings with small numbers of patients.

     We agree and accept your suggestion.

5.   Materials and Methods: (Lines 75-76) Chronic stroke is often defined as being 6 or 12 months post-stroke. Please provide rationale for requiring patients to be specifically 2-5 years post-stroke for this study.

     The boundaries of these phases are not clearly defined, as we can find some variations in the literature. The chronic phase can start at three, six or another moment according to the adopted criteria. Nevertheless, even starting at three months, the chronic phase ends with the end of life. Even without a rehabilitation treatment, several changes can occur in the first one-year post-stroke, as the type and severity of aphasia, sites and intensity of spasticity, voluntary control, pain etc [1,2]. Two-years post-stroke patients are in the maintenance phase, when even minimal positive modifications are possible but not expected, while a lack of specialized treatment may imply in negative changes. Since patients ranging 2-5 years post-stroke are not expected to improve movement, sensibility or pain level by themselves or due to the natural course of a stroke, our analysis and results could be strengthened by this criterium.

6.    (Lines 99-100) Please provide rationale for stimulating at 90% of resting motor threshold, as this intensity of stimulation is often considered under-powered.

     First, high-frequency rTMS tend to cause more discomfort with supra-threshold stimulation. Since the RMT for each participant ranged from 40 to 50% MSO, intensities of 100-120% could be painful for them. The recommendation is to use the lower intensity able to produce an effect in the target area, and 90% maximum stimulator output (MSO) was enough, agreeing with several studies that stimulated at 10 Hz with 80-90% MSO [3,4]. Second, the stimulation of non-motor areas is still under evaluation, and so far there is not a final statement about the better parameters to non-motor areas [5]. Third, the consideration that stimulations below 100% MSO may be under-powered is relevant for stimulations at 1 Hz or below, but it does not fit to high-frequency stimulations [6].

Results: I strongly suggest including a figure showing imaging (MRI/CT) results for the patients that demonstrates lesion location and evidence of spared parietal lobe.

     We have included a new figure with two MRI or TC slices of each participant, named Figure 2, and we readjust the other figures numeration (lines 139-148).

7.   Discussion: (Lines 194-195) Do the authors have any speculations as to why hand and wrist gains were even greater at the two-month evaluation compared to immediately post-treatment for P3?

     The inhibitory stimulation makes some region reduce its activity and is employed to allow that another region may work better. The excitatory stimulation, however, makes some region increase its activity and this increase acts more actively upon its connexions, which makes more centres increase its activity, thus triggering a more responsive circuitry [7,8]. Since the gains are rapidly incorporated into the daily living, they act like facilitators for new gains, and the activated neural environment is aligned with this facilitatory effect. An inhibitory stimulation could bring some gains, but although this gain also can function as a facilitator for new gains, the central nervous system is not activated for these new changes.

(Line 218) The statement “Precisely for this reason…” seems illogical. Why does the low prevalence of this condition specifically warrant the use of an excitatory protocol?

     “Precisely for this reason” refers to the fact that this approach brings the rare possibility of stimulating a spared area very close to the lesioned area in stroke patients. Thus, considering that we are stimulating a spared area, we should inhibit it; considering that we are stimulating the stroke hemisphere, we should excite it. Since we believe that the excitatory stimulation may be more beneficial, we choose the excitatory mode of rTMS. But we agree that a new construction could prevent this mistake, and we change it to “Precisely for this possibility of studying” (line 267).

8.   (Lines 222-223) Again, the statement “Thus, the model justifies conducting the study even with few patients” does not seem to be validated by the accompanying text.

     We changed the phrase “thus, the model justifies conducting the study even with few patients” to “therefore, in our view, the possibility to influence a lesioned area through the stimulation of an ipsilateral spared area justifies conducting the study even with few patients” (lines 275-277).

9.   I suggest re-writing the section from Line 212-223 to improve the logical flow of your arguments. It may well be justified to conduct this study with few patients, but the arguments provided do not seem to support that conclusion as-written.

We re-wrote this section.

10.(Lines 240-243) Concluding that the heterogeneity of the observed results is a strength of the study seems like a stretch. I suggest re-wording. This sentence is rather confusingly worded in general. Perhaps tone down the language regarding the strength of a heterogeneous result and instead point out that these preliminary results suggest that this protocol has the potential to improve a range of impaired functions.

We thank your suggestion and we changed the final sentences.

References  

1.   Pedersen PM, Vinter K, Olsen TS. Aphasia after stroke: type, severity and prognosis. The Copenhagen aphasia study. Cerebrovasc Dis. 2004;17(1):35-43. DOI: 10.1159/000073896

2.   Opheim A, Danielsson A, Alt Murphy M, Persson HC, Sunnerhagen KS. Upper-limb spasticity during the first year after stroke: stroke arm longitudinal study at the University of Gothenburg. Am J Phys Med Rehabil. 2014;93(10):884-896. DOI: 10.1097/PHM.0000000000000157.

3.   Maizey L, Allen CPG, Dervinis M, Verbruggen F, Varnava A. Comparative incidence rates of mild adverse effects to transcranial magnetic stimulation. Clinical Neurophysiology. 2013;124(3),536-544. DOI: 10.1016/j.clinph.2012.07.024)

4.   Sasaki N, Mizutani S, Kakuda W, Abo M. Comparison of the effects of high- and low-frequency repetitive transcranial magnetic stimulation on upper limb hemiparesis in the early phase of stroke. J Stroke Cerebrovasc Dis. 2013;22(4):413-418. DOI: 10.1016/j.jstrokecerebrovasdis.2011.10.004

5.   Lefaucheur JP, André-Obadia N, Poulet E, Devanne H, Haffen E et al. Recommandations françaises sur l’utilisation de la stimulation magnétique transcrânienne répétitive (rTMS): Règles de sécurité et indications thérapeutiques. Neurophysiol Clin. 2011;41(5–6):221-295.

6.   Lefaucheur JP, André-Obadia N, Antal A, Ayache SS, Baeken C et al. Evidence-based guidelines on the therapeutic use of repetitive transcranial magnetic stimulation (rTMS). Clin Neurophysiol. 2014;125(11):2150-2206. DOI: 10.1016/j.clinph.2014.05.021.

7.   Speer, A.M.; Kimbrell, T.A.; Wassermann, E.M.; Repella J.D.; Willis, M.W.; Herscovitch, P.; Post, R.M. Opposite effects of high and low frequency rTMS on regional brain activity in depressed patients. Biol Psychiatry 2000;(48):1133-1141.

8.   Valiulis V, Gerulskis G, Dapšys K, Vištartaite G, Šiurkute A, Mačiulis V. Electrophysiological differences between high and low frequency rTMS protocols in depression treatment. Acta Neurobiol Exp (Wars). 2012;72(3):283-295.

Reviewer 2 Report

Feedback to Authors

The manuscript titled ‘Effects of Excitatory Transcranial Magnetic Stimulation of the Anterior Intraparietal Area in Chronic Stroke Patients’ describes a case series of four subject with chronic frontal stroke patients (with spared parietal function) of whom 3 receive P3 (site of stimulation)  rTMS (10Hz at 90% M1 Rest Threshold stimulation intensity) and one receives ‘sham’ (coil held over scalp with no stimulation applied). Of the 3 patients who receive the P3 site rTMS there appears to be some varying responses in the FMA scores, with little change in the subject who received no stimulation.

I think this is an interesting study, and is for the most part well written. There are no ethical concerns with this manuscript, and the reasons for conducting the study are well stated. My main concerns are due to the very low numbers, making this  a case series, yet the authors seem to be making claims that a case series design cannot support. I am also concerned that the authors claim they are stimulating the anterior intraparietal area, yet this is unknown due to the inherent limitations of the chosen methods (ie, they have stimulated over P3 of the 10-20 system which may or may not have excited the AIP area). However, these major concerns can be mitigated by re-writing the manuscript to reflect these limitation. If these concerns are met, I do believe this study would be of interest to the Brain Sciences readership.

Major Concerns

1) Misleading stimulation site claims. Throughout the manuscript you claim to be stimulating the Anterior Intraparietal (AIP) area, based on reference 30. This is somewhat misleading. Although Herwig et al (ref 30) state that stimulation over the P3 site according to the 10-20 EEG system resulted in their study with activation of the intraparietal sulcus, they also state that in about 90% of the subjects this site stimulates that distinct cortical region or within up to two adjacent Brodmann areas. This means you could have been stimulating completely different brain regions than the AIP area. I fully understand the practical nature of choosing the P3 site as the stimulation site, and that is not a problem, but it is misleading to the readership to call this AIP stimulation. Please change this throughout the manuscript to P3 site stimulation, because that is what you have done. This may or may not have stimulated the AIP area in your patients. Please re-write your entire paper from title, abstract, introduction, methods, results, discussion and conclusions to state your stimulated the P3 site of the 10-20 system. In your discussion you can ‘discuss’ that this is possibly the AIP, but also possibly not.

REF 30: Herwig U, Satrapi P, Schönfeldt-Lecuona C. Using the International 10-20 EEG System for Positioning of Transcranial Magnetic Stimulation. Brain Topography. 12 / 01 / 2003;16(2):95-99.

2) Over-claiming rTMS effects that this study design does not support. As this is clearly a case series your claims of a treatment effect are not supported by your study design. For example, in your abstract conclusion you claim ‘Conclusion: AIP excitatory rTMS increased the FMA scores for lower and upper limb function, showing a broader effect when compared to M1 stimulation.’ As you have not performed any M1 stimulation in your study you cannot make any claims about your findings compared to M1 stimulation. You also cannot claim to have stimulated AIP as noted above. You also cannot claim your data show that your P3 rTMS stimulation increased the FMA scores as this is a case series and not a fully powered RCT. The FMA score changes may have been due to the P3 rTMS or it may have been due to other effects such as a Hawthorne effect or natural variation. You can only claim an actual effect due to your intervention if you use a properly designed randomized clinical trail methodology. I understand the difficulties actually conducting such a study due to a low number of patients that have such specific strokes. Regardless you cannot make claims of an effect from an intervention in a case series of 3 treated subjects. You make similar claims in your conclusions. Again these must be revised. Please also re-write the discussion section in light of this design.

3) Sham patient. In the Methods, page 3, line 106 you note that your sham was simply holding the coil over the subjects head, without it making any sounds. How likely are you in having successfully ‘shamed’ your subject? Having conduced multiple TMS studies I doubt you would have fooled your subject with this. Holding the tip of the coil against the head and having the machine stimulate the air above the patients head does at least make it sound like something is being done. I therefore have great concerns about this being a good or successful ‘sham intervention’. The limitations of your sham intervention should be noted in a far more extensive ‘limitations’ section in your discussion. Also the fact that this ‘sham’ patient was not followed up with a third session does not allow for the potential to see whether this subject changed at the A3 recording session. There may have been natural changes or a Hawthorn effect that could have been comparable to the so-called effects from the rTMS in the other 3 patients. This should also be noted in the limitations section.

4) a significantly larger limitations sections is needed for this case series.

Minor Issues

1)      Title; The title must change. You are not sure you have stimulated AIP so this should not go in the title.

2)      Title: Please note in the title it’s a case series

3)      Title: please note its repetitive TMS in the title

4)      Abstract, line 15/16 please change stimulation site to P3 of the 10-20 system

5)      Abstract; rTMS and ROM have not been introduced before acronyms used.

6)      Introduction, lines 32-34; This statement is somewhat misleading as the literature on this topic is conflicting. Multiple studies clearly show that many patients with these conditions do not respond at all to rTMS. If you wish to portray that there are some studies that show promising results, you should also balance this with noting the many studies that have shown little or no effect in the same patient populations. This may be, as you note later on, due to stimulation parameters and site so will not weaken the reasons for your study.

7)      Methods, page 3, line 96; in the description re obtaining rest threshold over the motor cortex, it is stated that you used the C3 site to obtain this. However, this may not have been the optimal site for eliciting a MEP in FDI, which would have led to potentially vastly different rest thresholds. Why was the optimal site used?  

8)      Please provide the actual MSO used for each individual?

9)      Methods, page 3, line 100; please provide justification for using 10Hz stimulation rate

10)   Methods, page 3, line 100; please provide justification for stimulus duration

11)   Methods, page 3, line 100; please change the word ‘totalizing’ to ‘totaling’

12)   Methods, page 3, line 104; please describe the orientation of the coil better. What direction current flow did this setup produce? Is this optimized to activate neurons in a sulcus (i.e. your target area).

13)   Results, page 3, line 126 and line 128 you refer to a patient but have not included a number so we do not know which patient you are referring to. Please add in patient number.

14)   Figure 1; please include the so-called sham patient results.

15)   Figure 1; please note in the legend that the clinically important differences refers to motor function only. The changes in sensory function in P2 appear to fall into the range of ‘minimal clinically important differences’ thus this is misleading for the reader. Maybe remove that yellow shaded section for this subject as this subjects motor function did not change, and therefore you will not mislead the reader about the clinical relevance of the sensory changes for P2 (i.e. the middle column).

16)   Figure 2; please add in the findings of the so-called sham subject to this figure.

17)   Figure 2; please note in the legend of this figure that the clinical importance yellow highlight pertains to grasping ability, thus relates to motor function only, and not for the ROM changes for Patient 3. Otherwise this is misleading the reader.

18)   Discussion, page 4 and 5, line 155-157, regarding the aim of the study. You need to re-write this aim of the study, as this study was not designed to assess an affect from rTMs. It’s a case series. You can only observe changes in these individual cases that may or may not be due to the rTMS. Also as noted before you cannot claim you stimulated the AIP. You stimulated over P3 period.

19)   Discussion, page 5, line 164; please correct A7 to A3.

20)   Discussion, page 6, lines 239-244, these are strong claims that are not supported by your study design. Please re-write this section.  

21)   Conclusion, page 6, lines 246-248, these are strong claims that are not supported by your study design. Please re-write this section completely.  

Author Response

Reviewers report – Brain Science

Reviewer 2

The manuscript titled ‘Effects of Excitatory Transcranial Magnetic Stimulation of the Anterior Intraparietal Area in Chronic Stroke Patients’ describes a case series of four subject with chronic frontal stroke patients (with spared parietal function) of whom 3 receive P3 (site of stimulation)  rTMS (10Hz at 90% M1 Rest Threshold stimulation intensity) and one receives ‘sham’ (coil held over scalp with no stimulation applied). Of the 3 patients who receive the P3 site rTMS there appears to be some varying responses in the FMA scores, with little change in the subject who received no stimulation.

I think this is an interesting study, and is for the most part well written. There are no ethical concerns with this manuscript, and the reasons for conducting the study are well stated. My main concerns are due to the very low numbers, making this a case series, yet the authors seem to be making claims that a case series design cannot support. I am also concerned that the authors claim they are stimulating the anterior intraparietal area, yet this is unknown due to the inherent limitations of the chosen methods (ie, they have stimulated over P3 of the 10-20 system which may or may not have excited the AIP area). However, these major concerns can be mitigated by re-writing the manuscript to reflect these limitation. If these concerns are met, I do believe this study would be of interest to the Brain Sciences readership.

     We would like to thank you for you comments. We are sure that they have greatly contributed to the improvement of our article. We hope we have responded appropriately to all of them.

Major Concerns

1) Misleading stimulation site claims. Throughout the manuscript you claim to be stimulating the Anterior Intraparietal (AIP) area, based on reference 30. This is somewhat misleading. Although Herwig et al (ref 30) state that stimulation over the P3 site according to the 10-20 EEG system resulted in their study with activation of the intraparietal sulcus, they also state that in about 90% of the subjects this site stimulates that distinct cortical region or within up to two adjacent Brodmann areas. This means you could have been stimulating completely different brain regions than the AIP area. I fully understand the practical nature of choosing the P3 site as the stimulation site, and that is not a problem, but it is misleading to the readership to call this AIP stimulation. Please change this throughout the manuscript to P3 site stimulation, because that is what you have done. This may or may not have stimulated the AIP area in your patients. Please re-write your entire paper from title, abstract, introduction, methods, results, discussion and conclusions to state your stimulated the P3 site of the 10-20 system. In your discussion you can ‘discuss’ that this is possibly the AIP, but also possibly not.

REF 30: Herwig U, Satrapi P, Schönfeldt-Lecuona C. Using the International 10-20 EEG System for Positioning of Transcranial Magnetic Stimulation. Brain Topography. 12 / 01 / 2003;16(2):95-99.

     We agree with your consideration. We modified the stimulation site throughout the article. This modification also implied in some modification of the introduction and the discussion. We included a new image illustrating the Herwig et al.’s findings regarding the Brodmann areas achieved by the P3 stimulation.

2) Over-claiming rTMS effects that this study design does not support. As this is clearly a case series your claims of a treatment effect are not supported by your study design. For example, in your abstract conclusion you claim ‘Conclusion: AIP excitatory rTMS increased the FMA scores for lower and upper limb function, showing a broader effect when compared to M1 stimulation.’ As you have not performed any M1 stimulation in your study you cannot make any claims about your findings compared to M1 stimulation. You also cannot claim to have stimulated AIP as noted above. You also cannot claim your data show that your P3 rTMS stimulation increased the FMA scores as this is a case series and not a fully powered RCT. The FMA score changes may have been due to the P3 rTMS or it may have been due to other effects such as a Hawthorne effect or natural variation. You can only claim an actual effect due to your intervention if you use a properly designed randomized clinical trail methodology. I understand the difficulties actually conducting such a study due to a low number of patients that have such specific strokes. Regardless you cannot make claims of an effect from an intervention in a case series of 3 treated subjects. You make similar claims in your conclusions. Again these must be revised. Please also re-write the discussion section in light of this design.

     We thank your comprehension for the minimal number of participants in our study. Although the randomized clinical trial studies are the gold-standard for the evidence-based medicine, studies like this make their contribution to the science in several ways. We agree that we must present our findings taking into account its variety, the sample size and the possible variation relative to the stimulated area. In this way, we cannot claim a general effect to the group, although we can discuss the similarities and discrepancies. Therefore, we modified the text and re-wrote the discussion. But we do not agree that our FMA score changes may have been due to natural variation since our sample was composed of chronic stroke patients with at least two years of injury. The natural course of the stroke is well defined in the literature and do not include natural changes spontaneously occurring two years or more after the injury. Since we found FMA scores whose variation achieved minimal clinically important difference we cannot impute these results to natural variation. We reserve the discussion about the Hawthorne effect to the following questions.

3) Sham patient. In the Methods, page 3, line 106 you note that your sham was simply holding the coil over the subjects head, without it making any sounds. How likely are you in having successfully ‘shamed’ your subject? Having conduced multiple TMS studies I doubt you would have fooled your subject with this. Holding the tip of the coil against the head and having the machine stimulate the air above the patients head does at least make it sound like something is being done. I therefore have great concerns about this being a good or successful ‘sham intervention’. The limitations of your sham intervention should be noted in a far more extensive ‘limitations’ section in your discussion. Also the fact that this ‘sham’ patient was not followed up with a third session does not allow for the potential to see whether this subject changed at the A3 recording session. There may have been natural changes or a Hawthorn effect that could have been comparable to the so-called effects from the rTMS in the other 3 patients. This should also be noted in the limitations section.

     We apologize for our simplistic description of our proceedings, which we believe we can improve to give you more confidence in our method. The coil used in the sham treatment was not attached to the machine, so this coil really did not produce any sound. However, the active coil was still attached to the machine. The machine was set with the same parameters for the four participants, so the active coil produced the pulse sound very close to the patient head. According to Loo et al. [1], a perfect ‘sham’ coil must meet four criteria, three of them compared to the active coil: the same position in the scalp, same somesthetic sensation and same auditive artefact. Not generating any physiological effect on the brain cortex is the fourth criteria. Coils that not touch the scalp were used as a ‘sham’ technique, although the first criterium was not achieved. Altered positions, as to put the coil 45° or 90° relative to the scalp do not agree with the second criterium, nor sham coils with different weights and compounded materials. As Lefaucheur et al. [2] pointed out, an ideal sham treatment for TMS is still not available since the auditive-somatosensorial stimuli produced by the active coil can’t be completely reproduced even if the coil is improved with skin electrical stimulation. Aligned to the actual state-of-art of sham treatment techniques, our technique is efficient and adequate.

     We changed the phrase “To determine each participant’s resting motor threshold (RMT), the coil was positioned with the handle at a 45° angle to the anterior-posterior axis” (lines 94-95) to “to determine each participant’s resting motor threshold (RMT), the coil was positioned tangentially on the scalp with the handle directed upward and posteriorly at a 45° to the frontal plane, nearly parallel to the central sulcus” and the phrase “The coil was positioned with its 45° reward to the frontal plane. Participants lay down their side on a stretcher during stimulation with head supported for comfort and better positioning of the coil. The sham patient was equally positioned, but the coil was unattached to the stimulator” was changed to “The coil was positioned tangentially on the scalp with the handle pointing posteriorly to the base of the neck at 30° relative to the transverse plane. This position follows the positioning described by Koch et al. [3] to better achieve the anterior intraparietal area. Participants lay down their side on a stretcher during stimulation with head supported for comfort and better positioning of the coil. The sham patient was equally positioned, but the sham coil was unattached to the stimulator, while the active coil was kept near the sham coil to provide the sham auditive stimulation”. The question about the limitation section and the Hawthorne effect will be addressed in the following question.

4) a significantly larger limitations sections is needed for this case series.

     We agree that we need to discuss the likely variety of stimulated areas. We also believe that we need to address the fact that we don’t have a third evaluation of our sham patient. These modifications are found in the lines 262-365.

     However, the Hawthorne effect deserves further consideration. Basically, this effect describes the behaviour modifications that occur with the observed group exactly because this group is aware of this observation [4]. As McCambridge says [5], if the Hawthorne effect exists, it is highly contingent on task and context. In addition, the notions of performance evaluation, due reward and punishment are strongly related to this phenomenon. Nevertheless, the behaviour modifications are essentially possible modifications: increasing the execution speed of a task, remembering of clean the hands follow exiting a contaminated room, improving the behaviour in communitarian spaces [6].

     In our study, we evaluated our patients three times: before starting the treatment, immediately following the end of the treatment and two months after the end of the treatment. Fugl-Meyer Assessment (FMA) was conducted on these three occasions. It is a very extensive evaluation compounded by several subsections that evaluate passive, active and synergic movements, pain, sensibility, and range-of-motion. It is a comprehensive evaluation that was indicated as the primary instrument to evaluate stroke patients in intervention trials [7]. FMA is a very complex instrument, and patients hardly could memorize their performances in previous evaluations that take place weeks or months ago. In addition, the result of numerous FMA items is not understandable for the lay subject, which prevents him from predicting the best result to be demonstrated. These FMA aspects make it difficult to change behaviour based on the perceptions of the researcher and family members' expectations because they hinder the identification of the behaviour to be modified. They are measures of movement, sensitivity, nociceptive response, motor coordination. They are concrete measures obtained by a careful evaluation.

     But the main aspect of our research that makes it difficult to associate our findings from each patient with the Hawthorne effect is the very nature of the selected patients. Prior to the study, all patients had stable symptoms. It is well known that stroke patients with two to five years of injury do not alter the measures evaluated by FMA by self-suggestion or driven by the idealization of a researcher's expectation since stroke patients plateau along the first-year post-stroke [8].  

     Therefore, we do not believe that the Hawthorne effect could respond to any finding of our research.

Minor Issues

1)      Title; The title must change. You are not sure you have stimulated AIP so this should not go in the title.

     Done

2)      Title: Please note in the title it’s a case series

     Done

3)      Title: please note its repetitive TMS in the title

     Done

4)      Abstract, line 15/16 please change stimulation site to P3 of the 10-20 system

     Done

5)      Abstract; rTMS and ROM have not been introduced before acronyms used.

     We introduced the rTMS acronym and the expression range of motion.

6)      Introduction, lines 32-34; This statement is somewhat misleading as the literature on this topic is conflicting. Multiple studies clearly show that many patients with these conditions do not respond at all to rTMS. If you wish to portray that there are some studies that show promising results, you should also balance this with noting the many studies that have shown little or no effect in the same patient populations. This may be, as you note later on, due to stimulation parameters and site so will not weaken the reasons for your study.

     We believe this statement is not misleading since it is the introduction first paragraph, and the idea has just started to be developed at this point. Indicating that several studies have obtained promising results we also indicate that it is not the reality of all of them. But we add the phrase “instead the variety of results of TMS with this population requires further studies” just to assure the better understanding of our text. Our introduction presents the use of excitatory stimulation in the current literature and the chosen area for stimulation. Thus, we also changed the paragraph describing the chosen area for stimulation to agree with the P3 stimulation.

7)      Methods, page 3, line 96; in the description re obtaining rest threshold over the motor cortex, it is stated that you used the C3 site to obtain this. However, this may not have been the optimal site for eliciting a MEP in FDI, which would have led to potentially vastly different rest thresholds. Why was the optimal site used?

     The scientific literature describes different techniques to define the RMT according to the available apparatus. A neuronavigational system could be useful in our research both to better define the rTMS hotspot and to better define the TMS hotspot to elicit the MEP in FDI, and we hope this acquisition will soon materialize. Nevertheless, the methodology we have employed in this study to define the RMT is correct and we can find it employed in the scientific literature, as in the Maizey et al.’s study [9].

8)      Please provide the actual MSO used for each individual?

     C1 = 40% à 90% (40) = 36% MSO

     C2 = 50% à 90% (50) = 45% MSO

     C3 = 45% à 90% (45) = 41% MSO

     S1 = 50% à 90% (50) = 45% MSO

9)      Methods, page 3, line 100; please provide justification for using 10Hz stimulation rate

     The excitatory stimulation is less used than the inhibitory, and few articles have compared the different frequencies normally employed. Stimulations at 10 Hz and 20 Hz are the most used. Taking into account that few or no differences were described in studies comparing these two stimulation frequencies, Lefaucheur et al. [2] advise giving preference to 10 Hz stimulation. Therefore, we included the phrase: The 10 Hz frequency was chosen according to international TMS guidelines, which advise that the 10 Hz frequency must be preferably chosen relative to 20 Hz, 15 Hz and 5 Hz”.

10)   Methods, page 3, line 100; please provide justification for stimulus duration

     Rossi et al. [10] have indicated that the maximum safety duration for train stimulation at 10 Hz ranging from 90% to 110% RMT is 5 seconds. We treated for 20 minutes with 25 seconds interval and 40 trains, thus each train lasted for 5 seconds. Lefaucheur et al. [2] have indicated that inter-train intervals lasting 1000 ms or less are dangerous, and our time interval lasted for 1500 ms. However, they point out that these data refer to stimulation of the primary motor cortex and non-motor regions data still need to be validated. Thus, we decided to follow these parameters aiming to assure the maximal safety of our participants because they are the better parameter in the literature by now.

11)   Methods, page 3, line 100; please change the word ‘totalizing’ to ‘totaling’

     Done

12)   Methods, page 3, line 104; please describe the orientation of the coil better. What direction current flow did this setup produce? Is this optimized to activate neurons in a sulcus (i.e. your target area).

     We addressed this question in the question about the sham patient.

13)   Results, page 3, line 126 and line 128 you refer to a patient but have not included a number so we do not know which patient you are referring to. Please add in patient number.

     In line 101, we presented the three treated patients. Among the four subjects, C1 was the only female patient. In addition, the narrative discourse presents one patient at a time, and the only patient cited so far was the patient C1. Therefore, lines 126 and 128 refer to patient C1. In the same way, following presentations cite the patient number and continue with his gender till the next presentation.

14)   Figure 1; please include the so-called sham patient results.

     Done.

15)   Figure 1; please note in the legend that the clinically important differences refers to motor function only. The changes in sensory function in P2 appear to fall into the range of ‘minimal clinically important differences’ thus this is misleading for the reader. Maybe remove that yellow shaded section for this subject as this subjects motor function did not change, and therefore you will not mislead the reader about the clinical relevance of the sensory changes for P2 (i.e. the middle column).

     We thank you for this observation, and we substitute the yellowish band for an individual highlight, thus eliminating this confusion. We also changed the figure description.

16)   Figure 2; please add in the findings of the so-called sham subject to this figure.

     Done

17)   Figure 2; please note in the legend of this figure that the clinical importance yellow highlight pertains to grasping ability, thus relates to motor function only, and not for the ROM changes for Patient 3. Otherwise this is misleading the reader.

     We substitute the yellowish band for an individual highlight and improved the figure description.

18)   Discussion, page 4 and 5, line 155-157, regarding the aim of the study. You need to re-write this aim of the study, as this study was not designed to assess an affect from rTMs. It’s a case series. You can only observe changes in these individual cases that may or may not be due to the rTMS. Also as noted before you cannot claim you stimulated the AIP. You stimulated over P3 period.

     We re-wrote the objective of the study considering the sample size we had and making all adjusts related to the stimulation site.

19)   Discussion, page 5, line 164; please correct A7 to A3.

Done

20)   Discussion, page 6, lines 239-244, these are strong claims that are not supported by your study design. Please re-write this section.  

We re-wrote this section.

21)   Conclusion, page 6, lines 246-248, these are strong claims that are not supported by your study design. Please re-write this section completely.

We re-wrote this section.

References  

1.   Loo CK, Taylor JL, Gandevia SC, McDarmont BN, Mitchell PB, Sachdev PS. Transcranial magnetic stimulation (TMS) in controlled treatment studies: are some ‘‘sham’’ forms active? Biol Psychiatry 2000;47:325-331

2.   Lefaucheur JP, André-Obadia N, Poulet E, Devanne H, Haffen E et al. Recommandations françaises sur l’utilisation de la stimulation magnétique transcrânienne répétitive (rTMS): Règles de sécurité et indications thérapeutiques. Neurophysiol Clin. 2011;41(5–6):221-295.

3.   Koch G, Versace V, Bonnì S, Lupo F, Lo Gerfo E, Oliveri M, et al. Resonance of cortico-cortical connections of the motor system with the observation of goal directed grasping movements. Neuropsychologia. 2010;48(12):3513–3520. DOI: 10.1016/j.neuropsychologia.2010.07.037

4.   Fry DE. The Hawthorne effect revisited. Editorial. Dis Colon Rectum 2018; 61: 6–7. DOI: 10.1097/DCR.0000000000000928

5.   McCambridge J, Witton J, Elbournec DR. Systematic review of the Hawthorne effect: New concepts are needed to study research participation effects. J Clin Epidemiol. 2014; 67(3):267–277. DOI:  10.1016/j.jclinepi.2013.08.015

6.   McDonald EG, Smyth E, Smyth L, Lee TC. Hand hygiene "hall monitors": Leveraging the Hawthorne effect. Am J Infect Control. 2018;S0196-6553(17)31299-3. doi: 10.1016/j.ajic.2017.11.030.

7.   Bushnell C, Bettger JP, Cockroft KM, Mattke S, Nilsen DM, Piquado T, Skidmore ER, Wing K, Yenokyan G. Chronic stroke outcome measures for motor function intervention trials: expert panel recommendations. Circ. Cardiovasc. Qual. Outcomes 2015;8:S163:S169, DOI:10.1161/CIRCOUTCOMES.115.002098.

8.   Toschke AM, Tilling K, Cox AM, Rudd AG, Heuschmann PU, Wolfe CD. Patient-specific recovery patterns over time measured by dependence in activities of daily living after stroke and post-stroke care: the South London Stroke Register (SLSR). Eur J Neurol. 2010 Feb;17(2):219-25. DOI: 10.1111/j.1468-1331.2009.02774.x.

9.   Maizey L, Allen CP, Dervinis M, Verbruggen F, Varnava A, Kozlov M, Adams RC,Stokes M, Klemen J, Bungert A, Hounsell CA, Chambers CD. Comparative incidence rates of mild adverse effects to transcranial magnetic stimulation. Clin Neurophysiol. 2013 Mar;124(3):536-44. doi: 10.1016/j.clinph.2012.07.024.

10.Rossi S, Hallett M, Rossini PM, Pascual-Leone A; Safety of TMS Consensus Group. Safety, ethical considerations, and application guidelines for the use of transcranial magnetic stimulation in clinical practice and research. Clin Neurophysiol. 2009;120(12):2008-2039. DOI: 10.1016/j.clinph.2009.08.016

Round 2

Reviewer 2 Report

The authors have done a great job in amending this manuscript, and its figures. It is much improved! There is only one place where the authors are still over-claiming in the discussion:

Discussion, line 266-269; You claim; ‘Here we found that the excitatory stimulation of the anterior intraparietalP3 point area 263 improved the lower limb FMA sensory function score for the three treated patients in different values, 264 suggesting that the anterior intraparietal area would excitatory stimulation of P3 may facilitate the 265 sensory input. Lower limb sensory input. Two patients obtained the maximum score for the upper 266 limb prior the treatment and the third patient did not change his upper limb sensory function score 267 after the treatment, therefore we cannot evaluate if the excitatory stimulation of P3 may improve it 268 or not.’ Your study design cannot verify this regardless of what results you got. A natural limitation of this type of design (case series) is that other causes may have influenced the outcomes, such as natural variation or natural progression. Please be mindful of the limitations of a case series. Please reword this statement. Plus there are methodological issues that may explain the variation in observed responses.

Other than this the manuscript only needs some English/Grammar issues amended, i.e. Minor Issues as follows:

Abstract line 22; use third person, not ‘we’

Abstract line 27; I would recommend a change to the final concluding sentence. Your current sentence is not good English. For example you could say something like ‘This study suggests excitatory rTMS over P3 may be of use for some chronic stroke patients, but these findings need to be verified in a future clinical trial.”

Introduction, line 32; please include references for first half of sentence stating rTMS is widely studied tool for the treatment of post stroke patients.

Introduction, line 34-35; clarify which population you mean, ie. Do you mean post-stroke population?

Introduction, line 45-47, poor English. Possible alternative could be; “Based on these findings a positive effect may also be possible with excitatory rTMS in a post-stroke population that is not restricted to the model of inter-hemispheric imbalance.”

Introduction, line 50-52; you claim; “However, direct application to the primary motor cortex may restrict the excitatory rTMS effects to the stimulated neurons since the main output of the primary  cortex is directed to the muscles and not to other areas of the brain, thus reducing the effectiveness of excitatory stimulation.” Are you sure this statement is correct? Can you back it with references?

Introduction, line 68, you appear to have accidentally deleted ‘regions’ from this sentence.

Discussion, line 222; You claim; “Several studies have pointed the relevance of the sensory function motor performance after stroke.” This statement makes no sense? Please reword to clarity meaning.

Discussion, line 239, what do you mean with ‘basal tone’? please clarify

Please reword the entire Limitations section on line 272-288. There are multiple English and Grammar problems with these paragraphs. Please also note that you do not need to justify a case study design. This is a novel study and important and interesting to the readership of Brain Sciences. You only need to highlight the natural limitations of such a design.

Discussion line 304; swap ‘achieve’ with ‘activate’ and line 305 swap ‘achieved’ with ‘activated’

Discussion line 306, where you say ‘ which reduced the possibility of generalization’.  Using the word ‘generalization’ here does not really make sense. Im assuming you are trying to say something like: ‘reduces the possibility of knowing which exact brain regions are activated by P3 stimulation, thus which exact brain regions that may have caused the observed clinical changes.’

Discussion, line 314-315; poor English, please rewrite

Conclusions, line 326; I would recommend adding another two sentence such as “However, as this is a case series of only three patients and one sham, we caution readers when interpreting these findings. These encouraging findings need to be verified in a future study utilizing an appropriate clinical trial design.”